# The mutational landscape of *Staphylococcus aureus* during colonisation

Francesc Coll [1,2,3,13] ✉, Beth Blane[4,13], Katherine L. Bellis [3,4,13], Marta Matuszewska [4,5], Toska Wonfor[6,7], Dorota Jamrozy [3], Michelle S. Toleman[4], Joan A. Geoghegan [6,7,8], Julian Parkhill [5], Ruth C. Massey [9,10,11], Sharon J. Peacock [4,14] & Ewan M. Harrison [3,4,12,14] ✉

*Staphylococcus aureus* is an important human pathogen and a commensal of the human nose and skin. Survival and persistence during colonisation are likely major drivers of *S. aureus* evolution. Here we applied a genome-wide mutation enrichment approach to a genomic dataset of 3060 *S. aureus* colonization isolates from 791 individuals. Despite limited within-host genetic diversity, we observed an excess of protein-altering mutations in metabolic genes, in regulators of quorum-sensing (*agrA* and *agrC*) and in known antibiotic targets (*fusA, pbp2, dfrA* and *ileS*). We demonstrated the phenotypic effect of multiple adaptive mutations in vitro, including changes in haemolytic activity, antibiotic susceptibility, and metabolite utilisation. Nitrogen metabolism showed the strongest evidence of adaptation, with the assimilatory nitrite reductase (*nasD*) and urease (*ureG*) showing the highest mutational enrichment. We identified a *nasD* natural mutant with enhanced growth under urea as the sole nitrogen source. Inclusion of 4090 additional isolate genomes from 731 individuals revealed eight more genes including *sasA/sraP, darA/pstA*, and *rsbU* with signals of adaptive variation that warrant further characterisation. Our study provides a comprehensive picture of the heterogeneity of *S. aureus* adaptive changes during colonisation, and a robust methodological approach applicable to study *in host* adaptive evolution in other bacterial pathogens.

*Staphylococcus aureus* is an important pathogen but also a commensal bacterium and part of the human microbiota. The anterior nares (lower nostrils) are the primary reservoir of *S. aureus* in humans, although the bacterium may colonise other body sites including the skin, pharynx, axillae, perineum and the intestine[1,2]. Despite being a commensal, when the epithelial barrier breaks or the immune system becomes compromised, *S. aureus* can cause a variety of infections, ranging from superficial skin and soft-tissue infections to life-threatening invasive

[1]Applied Microbial Genomics Unit, Department of Molecular Basis of Disease, Institute of Biomedicine of Valencia (IBV-CSIC), Valencia, Spain. [2]Department of Infection Biology, Faculty of Infectious and Tropical Diseases, London School of Hygiene & Tropical Medicine, London, UK. [3]Parasites & Microbes Programme, Wellcome Sanger Institute, Hinxton, UK. [4]Department of Medicine, University of Cambridge, Cambridge, UK. [5]Department of Veterinary Medicine, University of Cambridge, Cambridge, UK. [6]Institute of Microbiology and Infection, University of Birmingham, Birmingham, UK. [7]Department of Microbes, Infection & Microbiomes, College of Medicine & Health, University of Birmingham, Birmingham, UK. [8]Department of Microbiology, Moyne Institute of Preventive Medicine, School of Genetics and Microbiology, Trinity College Dublin, Dublin, Ireland. [9]School of Cellular and Molecular Medicine, University of Bristol, Bristol, UK. [10]School of Microbiology, University College Cork, Cork, Ireland. [11]APC Microbiome Ireland, University College Cork, Cork, Ireland. [12]Department of Public Health and Primary Care, University of Cambridge, Cambridge, UK. [13]These authors contributed equally: Francesc Coll, Beth Blane, Katherine L. Bellis. [14]These authors jointly supervised this work: Sharon J Peacock, Ewan M Harrison. ✉e-mail: fcoll@ibv.csic.es; eh6@sanger.ac.uk

infections such as bacteraemia. Colonisation is an important risk factor for *S. aureus* infection[3,4], and it is frequently the strain already colonising an individual that causes the infection[5,6].

Whilst a few studies have sought to characterise the adaptive changes that *S. aureus* undergoes during colonisation[7,8], our understanding remains incomplete. *S. aureus* persistently colonises ~25% of adults, while others are either never, or only transiently colonised (intermittent carrier)[9]. The genome of *S. aureus* encodes a range of adhesion, immune evasion and antimicrobial resistance factors that, when expressed, allow the bacterium to rapidly adapt to the nasal environment[10–14]. In addition to changes in gene expression, mutations in the genome of *S. aureus* will also be selected during colonisation if beneficial for survival. This is supported by data from an experimental human challenge model in which persistent carriers preferentially select their own strain, suggesting that *S. aureus* is adapted to the conditions encountered in the colonised individual[15]. This likely represents adaption to one or more of: (a) competition with other microbes in the nasal microbiota[9]; (b) nutrient availability in nasal secretions[14]; (c) adaption to the host immune response and other physiological variation; (d) spatial variation within the nasal environment (epithelium vs. hair follicles)[16]; (e) environmental exposures; and (f) the presence of therapeutic antibiotics and disinfectants (likely more acute in the clinical setting)[17].

*S. aureus* readily transmits between individuals and strain replacement may take place in persistently colonised individuals[18], meaning that *S. aureus* strains face common selective pressures when adapting to a new individual. Mutations conferring an advantage are therefore expected to be enriched within the same genes, or groups of functionally related genes, across multiple *S. aureus* colonising strains. To test this hypothesis, we analysed the genomes of clonal *S. aureus* isolates sampled from the same individuals to identify evidence of adaptation in recently diverged populations of bacteria. A similar approach has recently been applied to investigate genetic changes that could promote, or be promoted by, invasive infection[5,8,19], or associated with persistent or relapsing *S. aureus* bacteraemia[20–22].

To differentiate potentially adaptive genetic changes from neutral background mutation, we applied a genome-wide mutation enrichment approach to identify loci in the *S. aureus* genome under parallel and convergent evolution that could represent potential signals of adaptation during colonisation. Our results show that despite limited genetic diversity among colonising isolates of the same individual, multiple genes and pathways show a clear mutational signal of adaptation.

## Results

### Defining within-host genetic diversity in colonising isolates
To investigate putative adaptive genetic changes in *S. aureus* during colonisation we compiled a genomic dataset from 3497 *S. aureus* colonisation isolates from ten independent studies[5,23–31], which included a median of 2 isolates (IQR 2 to 4) from 872 individuals (Supplementary Fig. 1, Supplementary Table 1, Supplementary Data 1). The final dataset, after excluding unrelated (non-clonal) isolates from the same host and poor-quality genomes, consisted of 3060 isolate genomes from 791 individuals, and included 1823 nasal isolates (59.6%), 926 isolates from multi-site screens (30.3%) and 311 isolates from other colonising sites (10.1%). Of these, 1412 isolate genomes (46.1%) were genotypically identified as methicillin-resistant *S. aureus* (MRSA) and the remaining 1648 (53.9%) as methicillin-susceptible (MSSA). The high frequency of MRSA is due to the inclusion of studies that only targeted and screened for MRSA (6 out of 10). Out of the 791 *S. aureus* carriers, 769 (97.2%) were sampled in healthcare settings or nursing homes, and the rest from healthy carriers in other community settings. Most individuals (*n* = 701, 88.6%) were sampled in the UK, followed by Singapore (*n* = 62, 7.8%), Thailand (*n* = 17, 2.1%) and Ireland (*n* = 11, 1.4%). In

terms of timing between samples, 556 individuals (70.3%) had their isolates collected within two months, 126 (15.9%) between two to six months, and 95 (12.0%) between six to twelve months, and 14 (1.8%) more than a year apart.

The genetic diversity between isolates colonising the same individual was low (Supplementary Fig. 2), measured either as the number of single nucleotide polymorphism (SNPs) in the core genome (median 1 SNPs, IQR 0 to 4) or the number of genetic variants (SNPs and small indels) across the whole genome (median 3 variants, IQR 1 to 8). Putative recombination events were detected in the bacterial genomes of 15% of individuals (*n* = 117/791), accounting for 23% of the overall mutation count (*n* = 1721/7577) and were predominantly located (*n* = 1367/1721, 80%) in three prophage recombination hotspots within the reference genome used (NCTC8325)[32] (Supplementary Fig. 3). This is consistent with previous studies reporting that most homologous recombination in the core genome of *S. aureus* can be found at or around mobile genetic elements (MGEs)[33].

### Genome-wide mutation enrichment analysis identifies evidence of adaptation
To identify loci in the *S. aureus* genome exhibiting evidence of parallel and convergent evolution that could represent potential signals of adaptation during colonisation, we applied a genome-wide mutation enrichment approach (Fig. 1). Using clonal isolates sampled from the same host, we quantified the number of protein-altering mutations (missense, nonsense and frame-shift mutations) within each protein coding sequence (CDS) that arose de novo during *S. aureus* colonisation. We then statistically tested whether this was higher than expected when compared to the rest of the genome after correction for multiple testing.

Out of 2326 CDS tested, only the genes encoding the accessory gene regulator A (*agrA*), the accessory gene regulator C (*agrC*) and the assimilatory nitrite reductase large subunit (*nasD*) showed a statistically significant (*p*-value < 0.05 after adjusting for multiple testing) enrichment of protein-altering mutations (Fig. 2A). Just below the genome-wide significance level were genes encoding known antibiotic targets: *fusA* encoding the target of fusidic acid[34]; *dfrA* encoding the target of trimethoprim[35]; and *pbp2*, which encodes a target of beta-lactams[36] (Fig. 2A). The finding of *agr* genes (*agrA* and *agrC*), known to be frequently mutated in *S. aureus* carriers[37,38], and that of known antibiotic targets demonstrated the feasibility of our approach in detecting putative adaptive mutations.

To broaden the search for signals of convergent evolution in groups of genes that are functionally related, we counted mutations among all genes belonging to the same transcription unit (operon)[39]. Out of 1166 operons tested, nine reached statistical significance for an excess of protein-altering mutations (Fig. 2B). These included three operons containing genes that reached statistically significance on their own: U1306 (*nasD*) and U1096 and U1095 (both containing *agrA* and *agrC*); and six additional operons containing CDS that did not reach statistical significance on their own: overlapping operons U605, U606 and U604, all containing the *ileS* gene; the U942 operon harbouring four riboflavin biosynthesis genes (*ribD*, *ribB*, *ribA* and *ribH*); the U254 operon containing genes involved in fatty acid metabolism (*vraA*, *vraB* and *vraC*); and the U331 operon which includes a single hypothetical protein (*SAOUHSC_00704*) (Supplementary Data 2).

At the highest functional level, we aggregated mutations within CDS of the same metabolic process, as defined by well-curated metabolic sub-modules in the *S. aureus* JE2 reference genome[40]. Out of 61 metabolic pathways tested, 11 reached statistical significance for an excess of protein-altering mutations, with 'nitrogen metabolism' and 'riboflavin biosynthesis' pathways being the top two metabolic processes affected (Fig. 2C), demonstrating a clear signal of selection on distinct metabolic processes.

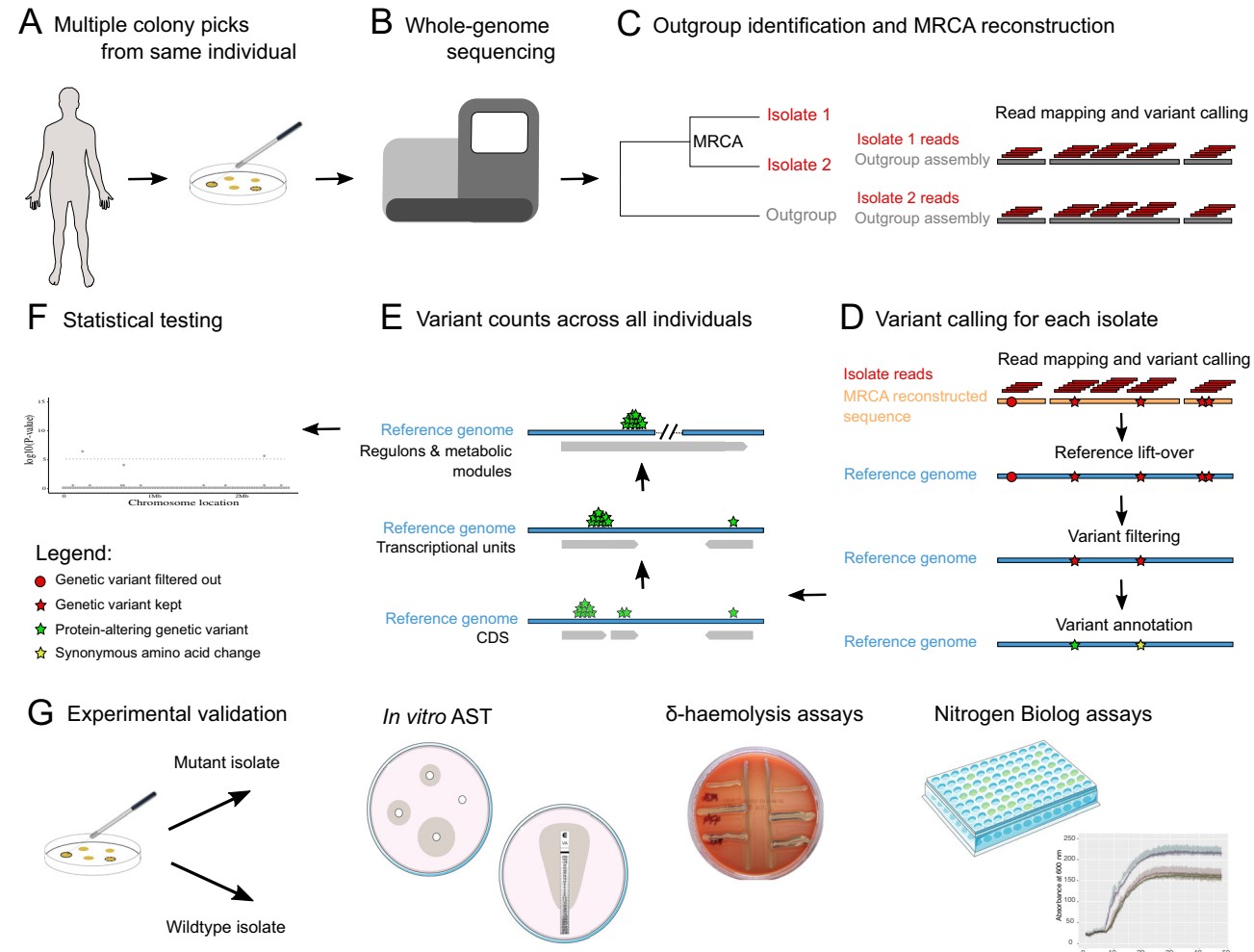

**Fig. 1 | Design of genomic analyses to detect potential signals of adaptation.**
**A** *S. aureus* colonies cultured from swabs taken from typical carriage sites of the same individual. (the human body silhouette icon was released into the public domain and obtained from https://commons.wikimedia.org/wiki/File:Human_body_silhouette.svg; the culture plate was created manually; and the inoculation loop obtained from https://bioicons.com/ under license CC0 1.0). **B** Multiple isolates are whole genome sequenced from the same individual (the sequencing machine icon was created manually). **C** A core-genome phylogeny is used to ensure isolates from the same host are clonal and to identify an appropriate outgroup. Isolate short reads are mapped to the outgroup assembly to call genetic variants. The sequence of the most recent common ancestor (MRCA) of all isolates from the same host is reconstructed. **D** The short reads of each isolate are mapped to the MRCA reconstructed sequence to call variants wherein the reference allele represents the ancestral allele and the alternative allele the evolved one. The coordinates of variants in a complete and well-annotated reference genome (Reference lift-

over) are determined. Variants on repetitive, low-complexity and phage regions are removed as well as those attributable to recombination (Variant filtering). In the last step, the effect of variants on genes is annotated (Variant annotation). **E** The number of protein-altering mutations are counted on protein-coding genes (CDS), transcriptional units (operons) and high-level functional units across all individuals. **F** Each functional unit is tested for an enrichment of protein-altering mutations compared to the rest of the genome. **G** The mutant isolate (with a putative adaptive mutation) and a closely related wildtype isolate obtained from the same individual are tested in vitro for antibiotic susceptibility (AST), delta-haemolytic activity, and growth under a variety of nitrogen sources to validate the phenotypic effect of putative adaptive mutations (the petri dish and multi-well plate icons by Servier https://smart.servier.com/ were obtained from https://bioicons.com/ under license CC-BY 3.0; the δ-haemolysis plate and the growth curves were obtained from this work).

## Nitrogen metabolic enzymes are enriched for mutations in colonising isolates

Nitrogen metabolism was the metabolic process most enriched by protein-altering mutations in colonisation isolates (Fig. 2C). *nasD* (also named *nirB*) was the third most frequently mutated gene (in a total of 14 individuals), only after *agrA* ($n = 19$) and *agrC* ($n = 20$). *nasD* encodes the large subunit of the assimilatory nitrite reductase, an enzyme responsible for reducing nitrite ($NO_2^-$) to ammonium, an early step in the fixation of nitrogen from inorganic forms (Fig. 3A). After *nasD*, the gene encoding the urease accessory protein UreG (*ureG*), was the second most mutated nitrogen metabolic enzyme (17th hit, Supplementary Data 2). Urease is a nickel-dependent metalloenzyme that

catalyses the hydrolysis of urea into ammonia ($NH_3$) and carbon dioxide ($CO_2$)[41].

Because urea is by far the most abundant organic substance in nasal secretions[14], we hypothesised that mutations in *nasD* and *ureG* could represent adaptations to the abundant availability of this nitrogen source. To investigate this, we first tested *nasD* and *ureG* transposon knockouts for their ability to grow under a variety of nitrogen sources. We observed rapid growth with amino acids like glycine, but slower growth with urea and ammonia as the primary nitrogen source, and even slower with nitrate and nitrite (Supplementary Fig. 4, Supplementary Data 3). Compared to the control strain (*comEB* transposon knock-out), the growth of the *nasD* knock-out was compromised in

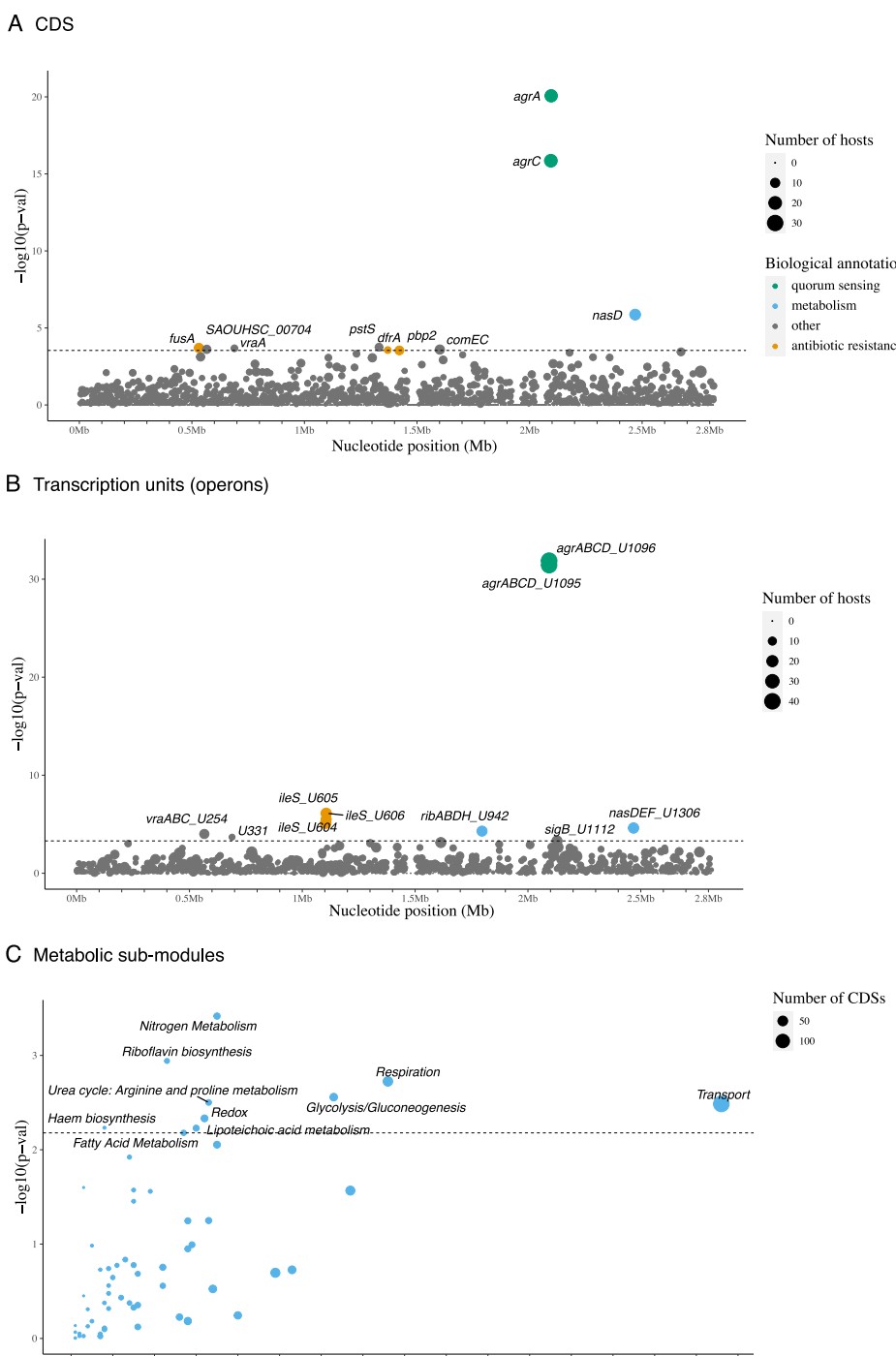

**Fig. 2 | Loci enriched for protein-altering mutations in colonising isolates.**
**A** Protein coding sequences (CDS) and (**B**) transcriptional units (operons) enriched for protein-altering mutations in colonising isolates of the same host. Each circle denotes a single locus, whose size is proportional to the number of hosts mutations arose independently from. Loci are placed at the x-axis based on their chromosome coordinates. The y-axis shows the uncorrected *p*-value resulting from a single-tailed Poisson test comparing the density of mutations per functional unit (CDS, transcription unit and metabolic sub-module) against the expected number of mutations, obtained from multiplying the genome-wide mutation count per bp by the gene length (see Methods). The dotted horizontal line represents the genome-wide statistical significance threshold. **C** Metabolic sub-modules enriched for protein-altering mutations in colonising isolates. In the x-axis, number of independent acquisitions of protein-altering mutations in different hosts across all protein-coding sequences (CDS) of the same metabolic sub-module. The number of CDS making up metabolic sub-modules is indicated with the size of each circle. In the y-axis, strength of statistic association shown by adjusted *p*-value. Mutations identified in a collection of 3060 *S. aureus* isolate genomes from 791 individuals (see Supplementary Table 1). Source data are provided as a Source Data file.

multiple nitrogen sources (Fig. 3C), including urea (growth rate 0.32 vs. 0.51, one-way ANOVA *p*-value < 0.01). Likewise, the growth rate of the *ureG* knock-out was significantly compromised with urea (0.25 vs. 0.51, one-way ANOVA *p*-value < 0.001, Supplementary Data 3),

highlighting the critical role of *ureG* in the utilisation of urea as the main nitrogen source for growth.

Next, we tested available colonising isolates with naturally occurring *nasD* mutations (Supplementary Table 2), and their

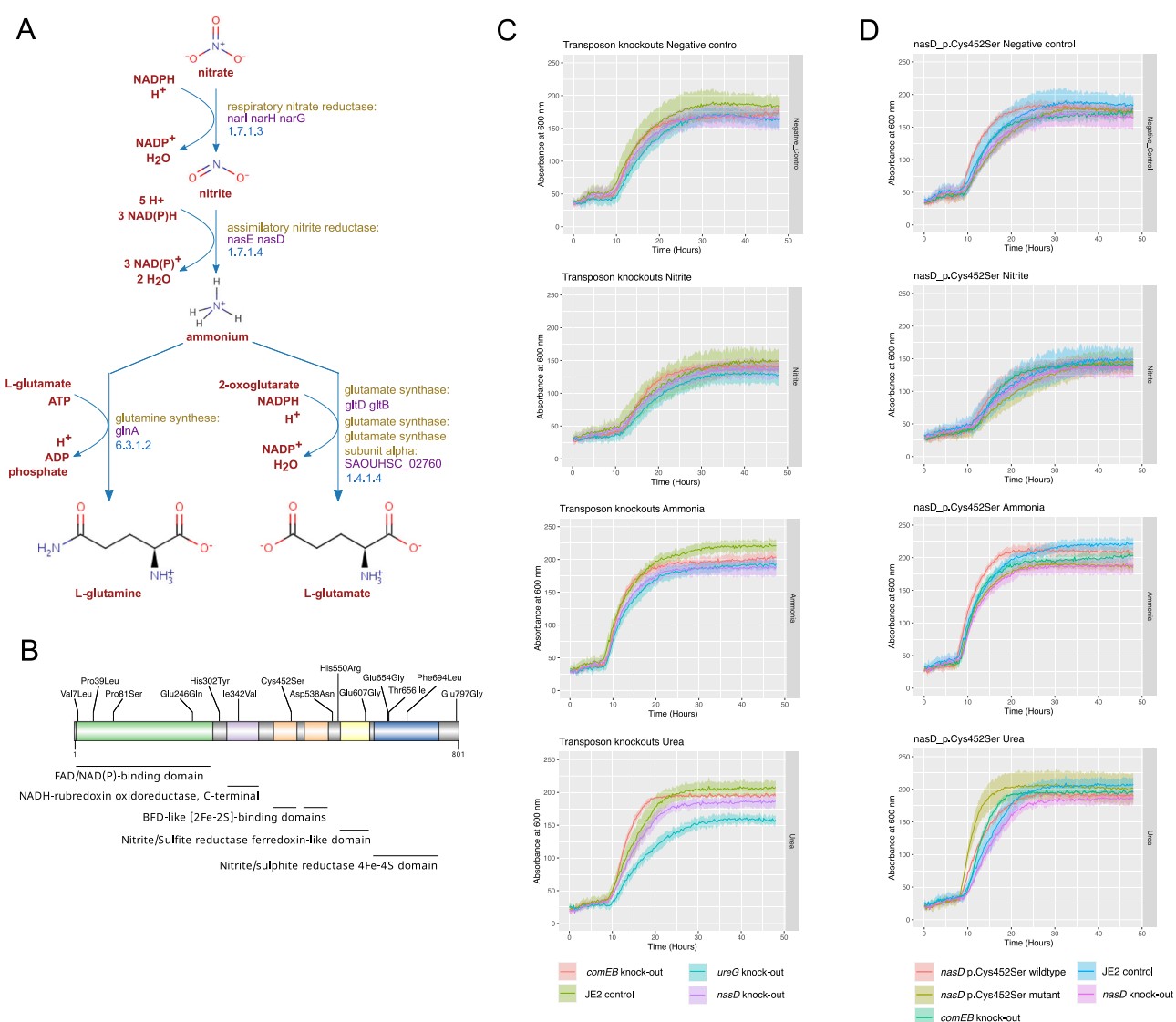

**Fig. 3 | Assimilatory nitrite reductase metabolic role, mutations, and nitrogen utilization of mutants. A** Role of the assimilatory nitrite reductase enzyme encoded by *nasD* in the nitrate assimilatory pathway of *S. aureus*. Adapted from BioCyc. **B** Location of missense mutations along NasD protein. Pfam protein domains are shown in distinct colours. **C** Growth curves of *S. aureus nasD/nirB* knock-out, *ureG* knock-out, *comEB* knock-out (control) and JE (control) strain under the following nitrogen sources: negative control well, nitrite, ammonia, and urea.

**D** Growth curves of *S. aureus nasD/nirB* p.Cys452Ser mutant, wildtype (i.e. quasi-isogenic isolate lacking the *nasD/nirB* mutation from the same host) and controls strains under the same nitrogen sources. Coloured lines represent mean OD600 calculated across three replicates, and shaded coloured regions the standard deviation. See Supplementary Data 5 for all mutations identified between colonising isolates of the same host. Source data are provided as a Source Data file.

corresponding closely related *nasD*-wildtype isolates from the same host (*n* = 3), for growth under the same nitrogen sources. Compared to the wildtype isolate, a Glu246Gln mutant (ST22) showed reduced growth under most nitrogen sources (Supplementary Fig. 5), including in the nitrogen negative control well, though the difference was most pronounced in urea, suggesting the fitness of this mutant was compromised relative to its wildtype. The Thr656Ile mutant (ST22) and wildtype both showed similar growth parameters across nitrogen sources, though the wildtype grew marginally better in urea than the mutant suggesting this mutation would be detrimental to growth in urea. Conversely, the Cys452Ser mutant (ST5) showed a statistically significant improvement in growth compared to its wildtype (in terms of a higher exponential growth rate: 0.64 vs 0.38, *p*-value < 0.001) in the presence of urea (Fig. 3D), compared to inorganic nitrogen sources. These results point to an adaptive effect of *nasD* Cys452Ser

mutation in the presence of urea. Interestingly, we also observed a strong effect of the strain's genetic background on growth, with ST5 isolates (Supplementary Fig. 5I–L) growing comparably as well as the transposon control strains (ST8), and ST22 isolates growing comparably worse.

## Adaptive mutations reveal well-known and unreported antibiotic resistance mutations

Our initial data suggested that the targets of antibiotics from distinct functional classes demonstrate a potential signal of adaptation (Fig. 2A). As such, we investigated whether mutations in these genes reduced susceptibility to their cognate antibiotics (Supplementary Fig. 6) by testing the antibiotic susceptibility of closely related clinical isolates that were mutant and wild-type pairs from the same individual (Fig. 1G). Mutations in *fusA* arose in 10 individuals. Out of the ten

missense variants (Supplementary Table 3), five had the exact amino acid changes previously reported to confer fusidic acid resistance (Val90Ile, Val90Ala, Pro404Leu)[42] or within the same codon (His457Arg) and were phenotypically resistant to fusidic acid. The other five isolates harbouring *fusA* missense variants were all susceptible to fusidic acid, ruling out an adaptive role of these mutations in fusidic acid resistance.

Five of the eight protein-altering mutations in *ileS* are known (Val588Phe and Val631Phe) or are in a codon (Gly593Ala) known to confer mupirocin resistance[42] and exhibited elevated MICs compared to the wildtype clonal isolate from the same individual (Supplementary Table 3). We confirmed the role of a not reported frameshift mutation (Ile473fs) in mupirocin resistance (E-test MIC 1024 µg/mL, breakpoint >12 µg/mL) and ruled out the effect of Gly591Ser (E-test MIC 0.5 µg/mL). Out of the five *S. aureus* isolates with missense variants in *dfrA*, three had amino acid changes reported to confer resistance to trimethoprim (His150Arg and two Phe99Tyr)[42]. The available isolate with Phe99Tyr was phenotypically resistant (MIC > = 16 µg/mL), but the isolate carrying His150Arg was not (MIC < = 0.5 µg/mL, zone diameter 27 mm), ruling out the role of this mutation in trimethoprim resistance in this particular strain background.

Missense mutations in *pbp2* were all located within the transglycosylase domain of PBP2 (Supplementary Fig. 6D), which is known to cooperate with PBP2A[43] to mediate beta-lactam resistance in MRSA. The three PBP2-mutated strains from available collections[23] were all ST22 (from phylogenetically distinct clades) MRSA (positive for *mecA*/PBP2a), but two were cefoxitin susceptible while retaining benzylpenicillin and oxacillin resistance (Supplementary Table 3). The corresponding PBP2-wildtype isolates from the same individual retain cefoxitin resistance, suggesting these mutations result in cefoxitin susceptibility.

We next investigated two sets of mutations putatively involved in glycopeptide resistance. First, *vraA*, a gene involved in fatty acid metabolism, was the seventh most mutated protein-coding gene (n = 8 individuals), and is downregulated in daptomycin tolerant strains[44]. Mutations in other genes involved in cell membrane lipid metabolism (e.g., *mprF/fmtC* or *vraT*, Supplementary Table 4)[45] are reported to reduce daptomycin susceptibility. Second, *pstS* a gene encoding a phosphate-binding protein, part of the ABC transporter complex PstSACB, was the fourth most frequently mutated protein-coding sequence (n = 7 individuals) (Fig. 2A). A point mutation in another phosphate transporter of *S. aureus* (*pitA*) increased daptomycin tolerance[46]. We hypothesised that protein-altering mutations in *vraA* and *pstS* could have a similar effect on daptomycin resistance. We determined daptomycin MICs and tolerance under a sub-inhibitory concentration of daptomycin (0.19 µg/mL) for the available *vraA*-mutated and *pstS*-mutated isolates (Supplementary Table 5), and with *pstS* and *vraA* loss-of-function (LOF) mutations from a larger collection (Supplementary Table 6). These results showed that neither the *pstS* or *vraA* mutations, or LOF mutations led to significant increases in daptomycin MIC, and only the mutant *pstS* p.Gln217* (mean AUC = 10.5, one-way ANOVA *p*-value < 0.01, Supplementary Data 4, Supplementary Fig. 7) showed increased daptomycin tolerance. The absence of improved growth of mutants relative to controls indicates that the primary driver of *pstS* and *vraA* mutations was not daptomycin tolerance and suggests these mutations could be also metabolic adaptions to fatty acid metabolism.

## Agr-inactivating mutations arise frequently in colonising isolates

The genes encoding the sensor kinase AgrC and the response regulator AgrA were, by far, the most frequently mutated genes (Fig. 2A), found in strains colonising 22 and 21 individuals, respectively (including one strain with both an AgrC and AgrA mutation). These genes belong to an operon encoding the accessory gene regulatory (Agr) system, a two-component quorum-sensing system that senses bacterial cell density and controls the expression of a number of important *S. aureus* virulence factors[47].

In AgrC, protein-altering mutations were concentrated in the histidine kinase (HK) domain (n = 16/20, Fig. 4a), potentially abrogating phosphorylation of AgrA. For AgrA, mutations were enriched in the DNA binding domain (n = 16/19, Fig. 4b), likely preventing the binding of phosphorylated AgrA to its cognate DNA binding region. We additionally inspected mutations in the *agr* intergenic region and found that four of the five mutations in this region fall close to the AgrA binding site of Promoter 2 (Fig. 4c). Altogether, these mutations likely abrogate expression of the Agr system by preventing the phosphorylation of AgrA or binding of phosphorylated AgrA to its cognate DNA binding region. To confirm this we tested putative *agr*-defective mutants, and their corresponding *agr*-wildtype isolate from the same host, for delta-haemolytic activity as a proxy for *agr* activity[48]. Given the large number of mutations to test (Supplementary Table 7), we selected 24 isolates from available collections[23] containing a representative mutation (i.e. missense, frameshift, stop gained and inframe indel) at each protein domain or intergenic region. As expected, the selected representative Agr-mutants were negative for delta-haemolytic activity, while their corresponding closely related wild-type isolates retained activity (Fig. 4b). Because *agr* mutants could also have arisen during laboratory passage[48], we sought to provide evidence of *agr* mutants arising *in host*. The identification of the same mutation in the same strain from independent swabs (i.e., taken at different time points) would point to a single origin of such mutation arising *in host*, as it is less likely for the same mutation to originate multiple independent times in the laboratory. Indeed, we found that all five individuals with multiple Agr-mutated isolates taken from independent swabs (Supplementary Table 7) carried the same Agr mutation.

Mutations that inactivate *agr* have been reported in previous studies, predominantly in *agrC* and *agrA* genes, both in healthy carriers[4,37,38] and from multiple types of infections[48], validating our approach to look for signals of adaptation. However, while some studies propose that *agr*-inactivating mutations arise more frequently in infected patients[5], others report similar frequencies in both infected and uninfected carriers[38]. To investigate this, we tested whether *agr* mutants were more common in carriers who had staphylococcal infections (206/791) compared to *S. aureus* uninfected carriers (497/791) (See Methods for definitions). We did not find this to be the case (7/206 vs. 10/497, *p*-value 0.17) after accounting for the number of sequenced isolates, genetic distance, collection, and clonal background as potential confounders (See Methods).

## Further putative adaptive mutations in an extended and larger dataset

Our original dataset was compiled in June 2019 (3060 isolates from 791 individuals). To strengthen our initial findings, we searched for newly published studies with multiple colonisation isolates sequenced per individual (up to June 2023), to increase the sample size of the dataset and the chances of detecting additional adaptive variation. We applied the same curation, genomic and QC methodological steps to keep only high-quality and clonal genomes of the same individual from colonisation sources. A total of 4090 additional isolate genomes (63.2% MRSA) obtained from 731 individuals and 15 different studies[49–63] were included (Supplementary Table 8, Supplementary Data 1). Most individuals were sampled in the USA (n = 309, 42.3%), followed by Singapore (n = 186, 25.4%), UK (n = 141, 19.3%), Japan (n = 30, 4.1%), Thailand (n = 26, 3.6%) and other countries (n = 39, 5.3%). Application of the genome-wide mutation enrichment approach to the combined dataset (7150 isolates from 1593 individuals) revealed even more genes reaching statistical significance for an excess of protein-altering

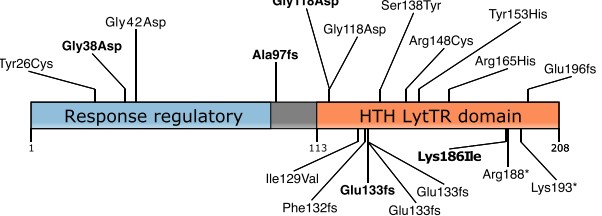

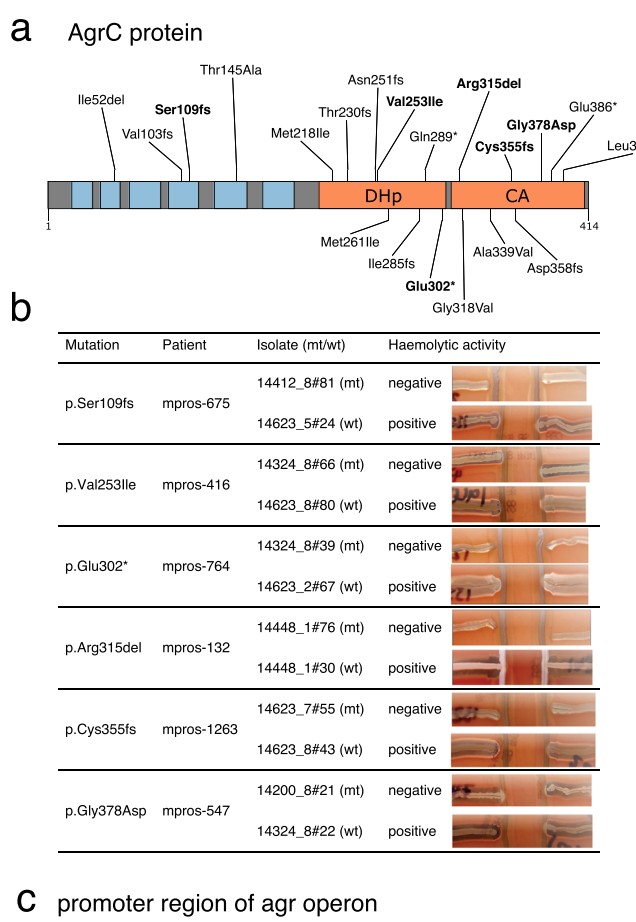

## b

| Mutation | Patient | Isolate (mt/wt) | Haemolytic activity | |
|---|---|---|---|---|
| p.Ser109fs | mpros-675 | 14412_8#81 (mt) | negative | |
| | | 14623_5#24 (wt) | positive | |
| p.Val253Ile | mpros-416 | 14324_8#66 (mt) | negative | |
| | | 14623_8#80 (wt) | positive | |
| p.Glu302* | mpros-764 | 14324_8#39 (mt) | negative | |
| | | 14623_2#67 (wt) | positive | |
| p.Arg315del | mpros-132 | 14448_1#76 (mt) | negative | |
| | | 14448_1#30 (wt) | positive | |
| p.Cys355fs | mpros-1263 | 14623_7#55 (mt) | negative | |
| | | 14623_8#43 (wt) | positive | |
| p.Gly378Asp | mpros-547 | 14200_8#21 (mt) | negative | |
| | | 14324_8#22 (wt) | positive | |

| Mutation | Patient | Isolate (wt/mt) | Haemolytic activity | |
|---|---|---|---|---|
| p.Gly38Asp | mpros-719 | 14355_2#54 (mt) | negative | |
| | mpros-719 | 14672_2#22 (wt) | positive | |
| p.Ala97fs | mpros-364 | 14412_8#75 (mt) | negative | |
| | mpros-364 | 8525_1#43 (wt) | positive | |
| p.Gly118Asp | mpros-1295 | 14412_8#14 (mt) | negative | |
| | mpros-1295 | 14623_7#61 (wt) | positive | |
| p.Glu133fs | mpros-254 | 14200_8#72 (mt) | negative | |
| | mpros-254 | 8447_7#92 (wt) | positive | |
| p.Lys186Ile | mpros-190 | 8447_7#11 (mt) | negative | |
| | mpros-190 | 14412_8#56 (wt) | positive | |

## c promoter region of agr operon

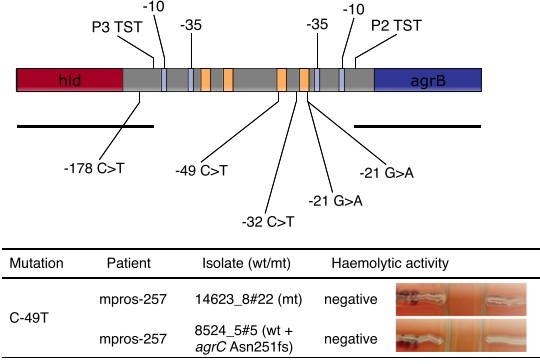

| Mutation | Patient | Isolate (wt/mt) | Haemolytic activity | |
|---|---|---|---|---|
| C-49T | mpros-257 | 14623_8#22 (mt) | negative | |
| | mpros-257 | 8524_5#5 (wt + agrC Asn251fs) | negative | |

**Fig. 4 | Mutations found in the accessory gene regulatory (Agr) system of colonizing strains. a** Protein-altering mutations in the protein domains of the sensor kinase AgrC and the response regulator AgrA. The N-terminal sensor domain of AgrC comprises six transmembrane domains (coloured in blue) and is connected to a conserved C-terminal histidine kinase (HK) domain (coloured in orange). The HK domain is made up of two subdomains: the dimerization and histidine phos-photransfer (DHp) subdomain and the catalytic and ATP-binding (CA) subdomain[110]. AgrA is comprised of a response regulatory domain (coloured in blue) and a DNA binding domain (coloured in orange). Isolates carrying mutations in bold were selected for haemolytic assays from available collections[23] to represent different types of mutations (i.e. missense, frameshift, stop gained and inframe indel) at each protein domain. **b** Haemolytic activities of *S. aureus* isolates on sheep blood agar (SBA) plates used to test the activity of the Agr system. For each mutation, two isolates from the same host were tested, one carrying a selected Agr

mutation (mutant) and a second isolate being wild type for the Agr system. A positive result is indicated by a widening of haemolysis seen in the region of RN4220. **c** Intergenic region containing agr promoters. The black horizonal lines represent the extent of transcript starting at the promoter 3 transcriptional start site (P3 TST), which encodes for RNAIII, and the transcript starting at promoter 2, which contains the whole *agrBDCA* coding region. Light blue boxes represent -10 and -35 boxes, whereas orange boxes the AgrA binding sites ("AgrA tandem repeats"). The only intergenic mutation carried by an available isolate (C-49T) yielded a negative haemolytic assay, as well as the isolate from the same host lacking this mutation, the latter attributable to a frameshift mutation in AgrC. Mutations identified in a collection of 3,060 *S. aureus* isolate genomes from 791 individuals (see Supplementary Table 1). See Supplementary Data 5 for all muta-tions identified between colonising isolates of the same host. Source data are provided as a Source Data file.

mutations (Fig. 5, Supplementary Fig. 8), including the ones originally identified (*agrA*, *agrC* and *nasD*) plus an extra eight genes. The latter included *darA/pstA* (which encodes a nitrogen regulatory protein), *sasA* (*S. aureus* surface protein A also known as SraP (serine-rich adhesin for binding to platelets involved in adhesion and invasion)[64,65],

*rsbU* (sigmaB regulation protein), and five genes yet to be functionally characterised (SAOUHSC_00704, SAOUHSC_00270, SAOUHSC_00621, SAOUHSC_02904 and SAOUHSC_00784).

The density of protein-altering mutations identified in SraP was highest within the L-lectin domain of the protein (Supplementary

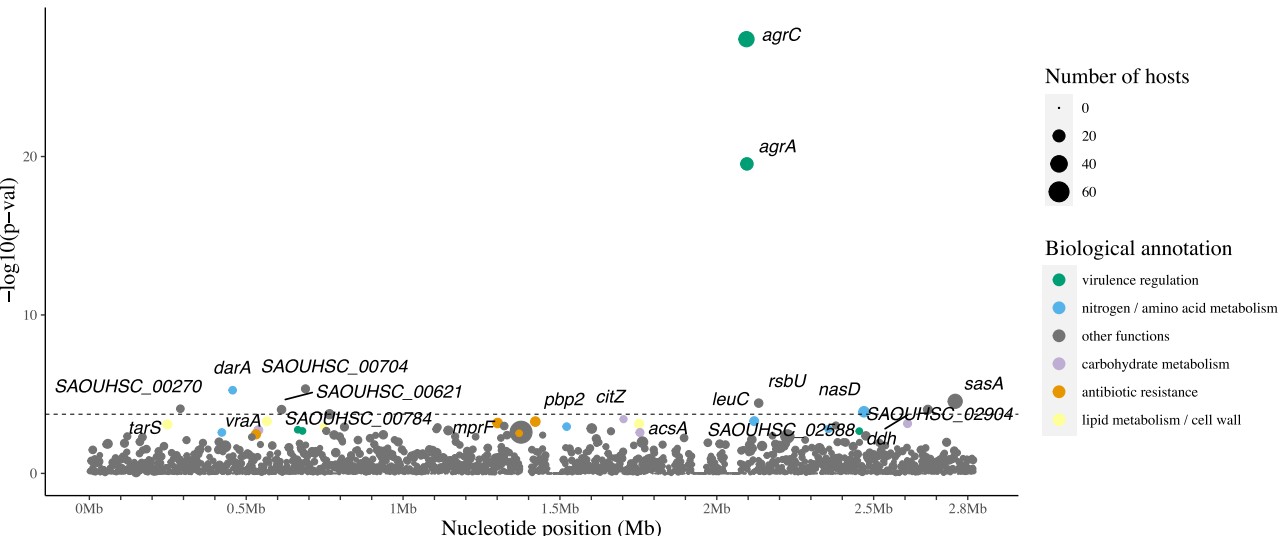

**A** Top 20 most significant CDS

**B** Statistically significant CDS (n=11)

| locus id | N mutations | gene name | product | p-value |
|---|---|---|---|---|
| SAOUHSC_02264 | 35 | *agrC* | accessory gene regulator protein C | $1.14 \times 10^{-24}$ |
| SAOUHSC_02265 | 22 | *agrA* | accessory gene regulator protein A | $4.23 \times 10^{-17}$ |
| **SAOUHSC_00704** | 8 | - | conserved hypothetical protein | $4.07 \times 10^{-3}$ |
| **SAOUHSC_00452** | 7 | *darA/pstA* | nitrogen regulatory protein | $4.07 \times 10^{-3}$ |
| **SAOUHSC_02990** | 31 | *sasA/sraP* | serine-rich adhesin for platelets | $1.63 \times 10^{-2}$ |
| **SAOUHSC_02301** | 9 | *rsbU* | sigmaB regulation protein RsbU | $1.77 \times 10^{-2}$ |
| **SAOUHSC_00270** | 8 | - | conserved hypothetical protein | 0.03 |
| **SAOUHSC_00621** | 9 | - | DMT family transporter | 0.03 |
| **SAOUHSC_02904** | 10 | - | Ferredoxin-NADP reductase | 0.03 |
| SAOUHSC_02684 | 15 | *nasD/nirB* | assimilatory nitrite reductase, large subunit | 0.03 |
| **SAOUHSC_00784** | 11 | - | Tetratricopeptide repeat protein | 0.04 |

**Fig. 5 | CDS enriched for protein-altering mutations in colonising isolates of the extended dataset. A** The top 20 most significant CDS are labelled on the plot. Each circle denotes a single locus, whose size is proportional to the number of hosts mutations arose independently from. Loci are placed at the x-axis based on their chromosome coordinates. The y-axis shows their uncorrected *p*-value resulting from a single-tailed Poisson test comparing the density of mutations in CDS against the expected number of mutations, obtained from multiplying the genome-wide mutation count per bp by the CDS length (see Methods). The dotted horizontal line represents the genome-wide statistical significance threshold. **B** Locus id and annotation of statistically significant CDS (*n* = 11) only. Locus IDs in bold indicate the genes that became statistically significant in the extended dataset. The second column shows the number of mutations originating in different hosts. The *p*-value presented in this table corresponds to the Benjamini-Hochberg corrected *p*-value. Mutations identified in a collection of 7150 *S. aureus* isolate genomes from 1593 individuals (See Supplementary tables 1 and 8). Source data are provided as a Source Data file.

Fig. 9A), a region which has previously been shown to bind to N-acetylneuraminic acid and mediates adherence to human lung epithelial cell lines in vitro[64]. Mapping the position of amino-acid substitutions arising from *sraP* mutations onto the crystal structure of the L-lectin domain revealed that N327 and G329 lie within 4 angstroms of the predicted N-acetylneuraminic acid binding site (Supplementary

Fig. 9B). Amino-acid substitutions at this site have the potential to directly impact ligand binding. Six additional amino-acid substitutions located at a further distance from the ligand binding site could influence N-acetylneuraminic acid binding by altering the protein fold, or could interfere with another as yet unknown function of SraP.

Genes encoding known antibiotic targets (*dfrA*, *fusA* and *pbp2*) remained among the top hits but below the genome-wide significance threshold. Among these was *mprF*, in which point mutations are known to confer daptomycin resistance. These results provide further evidence of the importance of nitrogen metabolism and identifies several uncharacterised genes likely to be critical for colonisation that warrant further experimental investigation.

## Discussion

In this study we have provided a comprehensive view of the mutational landscape shaped by selective pressures that *S. aureus* is exposed to during human colonisation. The frequency and type of genomic mutations that arise provide a record of adaptive changes that commensal *S. aureus* underwent in response to evolutionary pressures in the host and provide insights into the biology of *S. aureus* in its primary niche. We compared the genomes of isolates collected from the same host, across a large number of hosts, to detect loci under parallel and convergent evolution[66].

Our results provided indirect evidence of the ongoing metabolic adaptation of *S. aureus* during colonisation, with the strongest selective pressure being on nitrogen metabolism. We observed that nitrogen metabolic enzymes are often mutated in colonising isolates, specifically genes encoding sub-units of the assimilatory nitrite reductase (*nasD/nirB*) and urease (*ureG*), and a nitrogen regulatory protein (*darA/pstA*) in the extended dataset. Nitrite reduction can also be indicative of growth under anaerobic environments when nitrate ($NO_3^-$) and nitrite ($NO_2^-$) are used as terminal electron acceptors in place of $O_2$[67]. Indeed, genes related to dissimilatory nitrate and nitrite reduction are up-regulated under anaerobic conditions[68], when *nasD/nirB* serves to detoxify the nitrite that accumulates in nitrate-respiring cells[69]. Staphylococcal urease has also been implicated in adaptation to acid environments by ammonia production[41]. Therefore, it cannot be ruled out for *nasD/nirB* and *ureG* mutations could represent adaptations to anaerobic and acidic environments, respectively. Adaptions in nitrogen metabolic genes could be a response to distinct nitrogen source availabilities between human hosts, variation in the community structure of the nasal microbiota or between distinct sub-niches within the nasopharynx, therefore requiring metabolic adaptation of *S. aureus* strains when transmitting onto a new host or move to another anatomical niche. All in all, the specific phenotypic and physiological importance of variation in nitrogen metabolic genes is unclear and requires further characterisation.

Riboflavin biosynthesis was the second and eighth most mutated metabolic pathway in the initial and extended datasets, respectively, with *ribD* and *ribA* being the two most mutated genes in this pathway (Supplementary Data 2). Mutations in these genes might result in altered resistance to roseoflavin, an antimetabolite analogue of riboflavin that has antimicrobial properties[70], but we could not test this hypothesis as we lacked natural mutants of *ribD* and *ribA* from internal collections.

This is the first time that SraP/SasA, a member of the cell wall anchored (CWA) protein family found in Gram-positive bacteria, has been linked to nasal colonisation. *S. aureus* mutants deficient in two unrelated CWA proteins (ClfB and IsdA) were previously shown to be defective in colonising the nasal cavity of rodents due to a reduced ability to adhere to desquamated epithelial cells in the nares[71–73]. The interaction between SraP and N-acetylneuraminic acid mediates bacterial adherence to human lung epithelial cell lines in vitro[64] but a role in adherence to desquamated nasal epithelial cells has not been described. Future research will need to focus on determining the functional significance of SraP mutations occurring inside and outside of the L-lectin domain.

The targets of fusidic acid (elongation factor G), trimethoprim (dihydrofolate reductase), mupirocin (isoleucyl-tRNA synthetase) and

beta-lactams (penicillin-binding protein 2) showed a clear signal of adaptation as revealed by the independent emergence of mutations in the *S. aureus* isolates of multiple individuals. This most likely represents examples of directional selection, wherein *S. aureus* adapted to antimicrobial evolutionary pressures in vivo. The ancestral and evolved alleles could be determined using a closely related isolate sampled from a different individual as an outgroup (see Methods). This was supported by the identification of well-known resistance mutations in these genes, and concomitant reduced antibiotic susceptibility in isolates with these mutations, when compared to quasi 'isogenic' wild-type strains isolated from the same host. However, not all mutations detected in AMR loci were likely to be adaptive. This is exemplified by the characterisation of *fusA*, which had five mutations known to be involved in resistance leading to increases in MIC, and five never reported to cause resistance and not affecting fusidic acid susceptibility. It is therefore the excess of adaptive resistance-conferring mutations that increases the statistical significance of *fusA* and that of other AMR genes. We also identified unreported mutations suggesting that the full diversity of resistance mutations to these drugs is yet to be fully understood and warrants further study. The mutations identified in the transglycosylase domain of PBP2, two of which resulted in cefoxitin susceptibility, are consistent with the cooperation of this native PBP with the acquired PBP2A to mediate beta-lactam resistance in MRSA[43], and suggests that these might be compensatory mutations to optimise the function of the transglycosylase domain of PBP2. We hypothesised that protein-altering mutations in *vraA* and *pstS* could decrease susceptibility to daptomycin but we could not prove it. *pstS* mutations could be relevant to adaptation to phosphate limitation in colonising sites or in increasing resistance to nitric oxide stress by altering phosphate uptake[74]. The adaptive role of *vraA* and *acsA*, both encoding long chain fatty acid-CoA ligases, needs to be further investigated.

Our results support previous observations that *agr* genetic diversity is selected for during colonisation[38]. It has been proposed that a balance exists between wild-type and Agr-defective cells in the population, where the latter, termed as 'cheaters', benefit from the secretions of virulence factors by wild-type cells without having to produce the costly cooperative secretions[75]. However, in the context of colonisation, expression of the *agr* locus results in the down regulation of several surface proteins including cell wall secretory protein (IsaA)[76] and fibronectin binding protein B (FnBPB)[77], which are known to be involved in the attachment of *S. aureus* to cells in the nasal epithelium. Given the importance of these proteins to colonisation, it would be beneficial for *S. aureus* populations to maintain subpopulations of cells that are primed for attachment to a new host. Thus, mutations in *agr* most likely represent an example of balancing selection, where the bacterial population as a whole benefits from having both active and defective Agr systems, as opposed to a case of directional selection. Some caution is required however, as it is known that *agr* mutants arise frequently during the course of infections[5], and in vitro during post-isolation handling[48]. While we provide evidence of a subset of agr mutations arising *in host*, deep metagenomic sequencing of primary samples could be used in future studies to determine if mutations in *agr* are present before culture.

As previously proposed[8], adaptive antibiotic resistance mutations could result in other phenotypic effects (pleiotropic effects) or be instead the result of immune evasion adaptations. As such, *mprF* mutations reduce susceptibility to cationic antibiotics like daptomycin[78] but also to host defence cationic peptides produced by neutrophils[79], representing a potential immune evasion strategy. Similarly, mutations in *tarS*, which encodes the wall teichoic acid (WTA) β-glycosyltransferase and was frequently mutated in the extended dataset, could be adaptive in the context of immune evasion, as it has been shown that *tarS* knockout mutants escape host anti-WTA-IgG mediated opsonophagocytosis[80]. *tarS* variants could also be

influencing susceptibility to staphylococcal phages and beta-lactams antibiotics, as TarS-mediated WTA β-glycosylation is required for susceptibility to podoviruses[81] and methicillin resistance, respectively[82].

Compared to previous and similar studies reporting on *S. aureus* genetic adaptations, a recent study[8] also observed enrichments of mutations in *agrA*, *agrC*, *pbp2* and *mprF* between colonisation isolates of the same host. Mutations in genes encoding enzymes of the tricarboxylic acid cycle (TCA), such as *sucA*[8] and *fumC*[83], have also been detected frequently in skin-adapted strains. When investigating infection-specific adaptations, a high frequency of mutations in *agr* genes is often reported[5], as well as in genes involved in antibiotic resistance (e.g., *rpoB* and *rpsJ*)[8].

Our study has several limitations. First, the full genetic diversity of *S. aureus* in colonising sites was not captured as only a median of two sequenced colonies were available per individual. Second, we did not investigate changes in the gene content and large genetic rearrangements driven by movement of MGEs between isolate genomes of the same host, as this would require long-read sequencing. We did not explore the role of homologous recombination in adaption. Of note, 80% of mutations attributable to recombination fell within three prophage regions, consistent with previous studies showing that MGEs are hotspots of homologous recombination in the core genome of *S. aureus*[33]. These results point to a translocation and movement of Staphylococcal phages during colonization, the adaptive effect of which should be the focus of future studies. Third, we did not have metadata, such as antibiotic usage, or the specific site of colonisation (e.g., multi-site screens) for 30.3% of isolates. Fourth, most isolates came from studies of *S. aureus* colonisation in hospital patients with a bias towards MRSA strains, which may have incorporated a bias towards mutations selected by antibiotics or other therapies. Fifth, some of the mutational adaptions identified here could well represent adaptions of *S. aureus* strains to longer or extended periods of colonization, as those expected in persistent carriers, as these strains are more likely to be isolated over multiple sampling points and hence to be included in this study. Finally, generating and testing clean isogenic mutants would have provided stronger and more conclusive evidence about the role of individual mutations on the tested phenotypes. By increasing the overall sample size from ~3000 to ~7000 genomes we identified new genes significantly enriched for mutations including five currently uncharacterised genes and *sasA/sraP* which has not been previously been reported to be involved in colonisation, though it is known to mediate attachment to human cells[64]. This suggests that studies using even larger sample sizes have the potential to identify further new signatures of adaption.

Future work focused on pre-defined patient groups (healthy colonised individuals), narrowly defined infection types[53,84] with larger sample sizes and availability of host metadata will improve the identification of bacterial adaptive changes that promote survival in specific host niches and in vivo conditions; as well as defining strain/lineage- specific adaptations. Larger samples sizes will also allow us to determine which genes are essential for growth in different conditions, as shown by genes that are rarely inactivated.

While adaptation of clinical *S. aureus* strains during infection has been the focus of multiple recent studies[5,22,85–87], to our knowledge, this is the first comprehensive study to investigate adaptation of *S. aureus* populations during human colonisation. Our analysis has identified numerous metabolic pathways and genes likely critical to *S. aureus* colonisation that have not been previously reported and demonstrated the functional impact of these mutations. Our data now warrant detailed experimental investigations to further elucidate *S. aureus* biology during colonisation. Finally, it is likely that our approach can be applied to other bacterial species with similar success.

## Methods

### Strain collections and data curation

We identified available collections of *S. aureus* genomes with multiple carriage isolates sequenced from the same human individual (Supplementary Data 1)[5,23–31]. The NCBI Short Read Archive (SRA) was systematically queried on June 2019 to identify BioProjects that met the following criteria (Fig. 1): contained *S. aureus* genomic sequences, could be linked to a publication, included genomes of clinical isolates, clinical sources were known, multiple colonising isolates per host were sequenced, and host IDs were available. Only isolates from colonisation specimens were kept, that is, from multi-site screens[23,27,29,30] and typical colonising anatomical sites (nose[5,28,31], armpit, groin, perineum and throat)[24–26]. Colonised hosts were classified as symptomatic or asymptomatic carriers based on whether they had a *S. aureus* infection or not, respectively. In studies where clinical specimens were systematically collected from recruited cases[24,26,30], individuals were labelled as asymptomatic carriers unless having a clinical specimen collected. In other studies, carriers were all explicitly referred to as infected[5] or uninfected[30,31]. In one study, only the nasal carriage controls were kept, as were thus labelled as uninfected. In the rest of studies, no information was available to determine their *S. aureus* infection status[25,27,29], and were thus labelled as 'unknown'.

### Genomic analyses applied to all isolates

The Illumina short reads of all *S. aureus* genomes were validated using *fastqcheck* v1.1 (https://github.com/VertebrateResequencing/fastqcheck) and de novo assembled using Spades v.3.12.0[88] to create draft assemblies. These were then corrected using the bacterial assembly and improvement pipeline[89] to generate improved assemblies. All assemblies were evaluated using QUAST v5.0.1[90] and reads mapped back to de novo assemblies to investigate polymorphisms (indicative of mixed cultures) using Bowtie2 v1.2.2[91]. Kraken v2.1.2 was used to determine the proportion of *S. aureus* reads in raw fastq files. Low-quality genomes were excluded from further analysis applying the following thresholds: *S. aureus* reads <80%, N50 < 10000, contigs smaller than 1 kb contributing to >15% of the total assembly length, total assembly length outside of the median ± one standard deviation, or >1500 polymorphic sites.

Methicillin genotypic resistance was determined from improved assemblies using AMRFinderPlus[92] v3.11.11 (AMRFinder database v2023-08-08.2). Sequence types (STs) were derived from improved assemblies by extracting all seven *S. aureus* multi-locus sequence type (MLST) loci and comparing them to the PubMLST database (www.PubMLST.org)[93]. Clonal complexes (CCs) were derived from these allelic profiles, allowing up to two allele mismatches from the reference ST. The short reads of each isolate were mapped to the same reference genomes (CC22 HO 5096 0412 strain, accession number HE681097) using *SMALT* v0.7.6 (http://www.sanger.ac.uk/resources/software/smalt/), whole-genome alignments were created by calling nucleotide alleles along the reference genome using *SAMtools* and *bcftools* v0.1.19[94]. We kept the portion of the reference genome corresponding to the *S. aureus* core genome in whole genome alignments to calculate core-genome pairwise SNP distances using *pairsnp* v0.0.1 (https://github.com/gtonkinhill/pairsnp). The core genome of *S. aureus*[95] was derived from an independent, genetically and geographically diverse collection of 800 *S. aureus* isolates genomes from multiple host species[96] using *Roary*[97] v3.11.1 with default settings. Core-genome alignments were used to construct a maximum likelihood phylogeny for each clonal complex using *RAxML* v8.2.8[98] with 100 bootstraps.

### Genomic analyses applied to isolates of the same host

To avoid comparing the genomes of divergent strains from the same individual, only clonal isolates were kept for further analyses. Clonality was ruled out if isolates belonged to different clonal complexes or to

the same clonal complex separated by more than 100 SNPs. Clonality was ruled in if isolates differed by less than the maximum within-host diversify previously reported (40 SNPs)[99]. Clonality was investigated for the remaining isolates pairs (differing between 40 to 100 SNPs) by making sure they all clustered within the same monophyletic clade in the phylogenetic tree.

The nucleotide sequence of the most recent common ancestor (MRCA) of all isolates of the same host was reconstructed first. To do this, we used the maximum likelihood phylogenies to identify, for each individual, the most closely related isolate sampled from a different individual that could be used as an outgroup. We used the de novo assembly of this outgroup isolate as a reference genome to map the short reads of each isolate, call genetic variants (SNPs and small indels) using *Snippy* v4.3.3 (https://github.com/tseemann/snippy), and build within-host phylogenies using *RAxML* phylogeny and rooted on the outgroup. The ancestral allele of all genetic variants at the internal node representing the MRCA of all isolates of the same host was reconstructed using *PastML* v1.9.20[100]. This reconstructed ancestral sequence was used as the ultimate reference genome to call genetic variants (SNPs and small indels). This pipeline was implemented in four python scripts (identify_host_ancestral_isolate.step1.py to identify_host_ancestral_isolate.step4.py) available at https://github.com/francesccoll/staph-adaptive-mutations.

As variants were called in a different reference genome for each individual's *S. aureus* strain, they had to be transferred to the same reference genome to allow comparison and annotation of genetic variants from all individuals' strains. We modified an already published script (*insert_variants.pl*)[66] to find the genome coordinates of variants in the NCTC8325 (GenBank accession number NC_007795.1) and JE2 (NZ_CP020619.1) reference genomes. This script takes a 200-bp window around each variant in one reference (assembly) and finds the coordinates of this sequence in a new reference using *BLASTN*[101] and *bcftools* v1.9[94]. Because of this requirement, variants at the edge of contigs (200 bp) were filtered out. The script was modified to keep the single best blast hit of each variant, meaning that variants with window sequences mapping to repetitive regions of the reference genome were removed. Variants in repetitive regions, detected by running Blastn v2.8.1+ on the reference genome against itself, and variants in regions of low complexity, as detected by *dustmasker* v1.0.0[102] using default settings, were also filtered out. The final set of high-quality variants were annotated using *SnpEff* v4.3[103] in both the NCTC8325 and JE2 reference genomes. Breseq v0.39.0[104] was run to identify mutations and large indels between the pairs of related isolates from the same individual that were tested in vitro (Supplementary Data 5).

### Genome-wide mutation enrichment analysis
To scan for potential adaptive genetic changes recurrent across multiple individuals, we counted the number of functional mutations (i.e., those annotated as having HIGH or MODERATE annotation impact by *SnpEff*) in well-annotated functional loci across all individuals. Before that, putative recombination events, identified as variants clustered within a 1000-bp window in isolate genomes of the same host, were filtered out to avoid inflating mutation counts. When more than two isolates from the same host were available, we made sure the same mutations, identified in multiple case-control pairs of the same host, were counted only once.

We aggregated protein-altering mutations within different functional units. At the lowest level, we counted mutations within each protein coding sequence (CDS). To increase the power of detecting adaptive mutations in groups of genes that are functionally related, we aggregated mutations within transcription units (operons). The coordinates of transcription start and termination sites in the NCTC8325 reference genome were extracted from a study that comprehensively characterised the transcriptional response of *S. aureus* across a wide range of experimental conditions[39,105]. To our knowledge, this is the

best characterised reconstruction of transcriptional units in *S. aureus*. At the highest functional level, we aggregated mutations within CDS of the same metabolic process, as defined by well-curated metabolic sub-modules in the JE2 reference genome[40].

We tested each functional unit (CDS, transcription unit and metabolic sub-module) for an excess of protein-altering (functional) mutations compared to the rest of the genome, considering the length of CDS, or cumulative length of CDS if testing high-order functional units involving multiple CDS. To do this, we performed a single-tailed Poisson test using the genome-wide mutation count per bp multiplied by the gene length as the expected number of mutations as previously implemented[66]. Annotated features shorter than 300 bp long were not tested. *P* values were corrected for multiple testing using a Benjamini & Hochberg correction using the total number of functional units in the genome as the number of tests. We chose a significance level of 0.05 and reported hits with an adjusted *P* value below this value, unless otherwise stated.

### Other statistical analyses
We tested whether the presence of agr mutants, defined as isolates with protein-altering mutations in either *agrA* or *agrC*, was affected by hosts having an *S. aureus* infection (infection status). We fitted a binomial generalized linear model (GLM) using the presence of agr mutants as the binary response variable and *S. aureus* infection status as a binary predictor variable. We additionally included the number of sequenced isolates per host, genetic distance of these (expressed as the number of core-genome SNPs), collection and clonal background (clonal complex) as covariates to control for the effect of these potential confounders. This was implemented using the "glm" function (family binomial) in the base package within the statistical programming environment R version 3.4.1[106]. The only predictors that increased the odds of detecting agr mutants were the number of sequenced isolates per host (odds ratio 1.20, 1.10 to 1.34 95% confidence interval, *p*-value < 0.001) and their genetic distance (odds ratio 1.06, 95% confidence interval 1.01 to 1.11, *p*-value < 0.05).

### In vitro antibiotic susceptibility testing
Isolates from frozen stocks were grown overnight on Columbia blood agar (CBA, Oxoid, UK) at 37 °C. Fusidic acid or trimethoprim susceptibility testing was performed using disc (Oxoid, UK) diffusion as per EUCAST recommendations[107]. Minimum inhibitory concentration (MIC) testing was performed for daptomycin, vancomycin and mupirocin. A loopful of the isolate added to phosphate buffered saline (PBS), adjusted to 0.5 McFarland, then a thin layer spread evenly on a Muller Hinton agar plate (Oxoid, UK). An antimicrobial gradient strip (Biomerieux, France) was carefully placed, then the plate incubated overnight at 37 °C. The MIC was interpreted as the value on the strip above the point where growth stops.

### Biolog experiments
Isolates from frozen stocks were plated on to Lysogeny broth (LB) agar and grown overnight at 37 °C. For the transposon knock-out strains, obtained from Nebraska transposon mutant library, the plates included 5 ug/ml erythromycin. A damp swab was used to take sufficient colonies to create three 81% (+/- 2%) turbidity solutions for each strain in 20 ml PBS. 1.28 ml of each turbid solution was added to 14.83 ml 1.2x IF0a (77268, Biolog), redox dye H (74228, Biolog), and PM3 Gram Positive Additive (made as described by the Biolog protocol). Each well of a PM3 plate (12121, Biolog) was inoculated with 150 μl of this solution. The inoculated plates were run on the Omnilog (Biolog) for 48 h at 37 °C. Readings were taken every 15 min.

### Delta-haemolysis experiments
The δ-haemolysis assay was performed as previously described[48]. A thin streak of *S. aureus* strain RN4220 was placed down the centre of a

sheep blood agar plate. A thin streak of the test strain was placed horizontally up to, but not touching, RN4220. Test strains were tested in duplicate. Plates were incubated at 37 °C for 18 h, then at 4 °C for 6 h. Enhanced lysis by the test strain in the area near to RN4220 was an indicator of δ-haemolysis production.

## Growth curves

Test isolates were grown overnight at 37 °C in tryptic soy broth (TSB) with 5 ul/ml erythromycin (transposons) or TSB alone (non-transposons). The overnight cultures were then diluted 1/1000 in minimal media (1× M9 salts, 2 mM MgSO4, 0.1 mM CaCl2, 1% glucose, 1% casaminoacids, 1 mM thiamine hydrochloride and 0.05 mM nicotinamide) with 0.095 ug/ml daptomycin. 300ul was added to a 96-well plate, then placed on a FluoStar Omega (BMG Labtech, Germany) for 24 h incubation with shaking. Optical density measurement at $OD_{600}$ was taken every 30 min, and standard curves produced. Each isolate was tested in biological and measurement triplicate.

The R scripts used to process raw growth data, plot growth curves, fit growth curves and compare growth parameters are available on GitHub (https://github.com/francesccoll/staph-adaptive-mutations/tree/main/growth_curves). Raw growth data (i.e. absorbance values at different time points) was processed with script prepare_growth_curves_data.R. Mean OD600 values and 95% confidence limits around the mean were plotted using ggplot2[108] functions in script plot_growth_curves.R. Growth curves were fitted with Growthcurver[109] and growth parameters (growth rate and area under the curve) extracted using script fit_and_plot_growth_curves.R. Due to the prolonged lag phase in curves obtained under daptomycin exposure, these curves were fitted after 7 h. We fitted logistic curves to each replicate ($n = 9$) using Growthcurver package in R and extracted the growth rate and area under the logistic curve from fitted curves. These growth parameters were compared between isolates/strains (e.g. mutant vs. wildtype) using a one-way ANOVA to determine whether there were any statistically significant differences between the means (across replicates) of growth parameters between isolates (script: compare_growth_parameters.R).

## Reporting summary

Further information on research design is available in the Nature Portfolio Reporting Summary linked to this article.

## Data availability

The whole genome sequences of the initial and additional isolate collections used in this study are available on European Nucleotide Archive (ENA) under the accessions listed in Supplementary Data 1, which also includes isolate metadata. The initial dataset includes isolate genomes obtained from BioProject accessions PRJEB11177, PRJEB20148, PRJEB2076, PRJEB2655, PRJEB2756, PRJEB3174, PRJEB4141, PRJEB7654, PRJEB9390, PRJNA270998, PRJNA324190, and PRJNA369475; and the additional datasets from PRJDB5246, PRJEB2076, PRJEB28206, PRJEB33854, PRJEB40888, PRJEB4140, PRJEB43023, PRJEB9390, PRJNA530184, PRJNA587530, PRJNA590514, PRJNA595570, PRJNA638400, PRJNA685142, PRJNA715375, PRJNA715649, PRJNA816913, and PRJNA918392. The full list of protein-coding regions, transcriptional units and metabolic processes enriched by protein-altering mutations can be found in Supplementary Data 2. Supplementary Data 3 and 4 include the data of bacterial growth curves. Source data are provided with this paper.

## Code availability

All scripts necessary to run the described analyses are available on GitHub (https://github.com/francesccoll/staph-adaptive-mutations) and Zenodo (https://zenodo.org/records/14168960, https://doi.org/10.5281/zenodo.14168960)

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

## Acknowledgements

This publication presents independent research supported by Wellcome grants 201344/Z/16/Z and 204928/Z/16/Z awarded to Francesc Coll. This publication was also supported by the Health Innovation Challenge Fund (WT098600, HICF-T5-342), a parallel funding partnership between the Department of Health and Wellcome Trust. E.M.H. was supported by a UK Research and Innovation (UKRI) Fellowship: MR/S00291X/1. MST was a Wellcome Trust Clinical PhD Fellow at the University of Cambridge. RCM is funded by a Science Foundation Ireland Frontier for the Future Program Award (reference: 21/FFP-A/9704), and a Wellcome Trust Investigator Award (ref: 212258/Z/18/Z). This publication was also supported by Wellcome Grant reference: 220540/Z/20/A, 'Wellcome Sanger Institute Quinquennial Review 2021-2026' and Wellcome Collaborative Award in Science: 211864/Z/18/Z. This research was supported by the NIHR Cambridge Biomedical Research Centre (NIHR203312*). The views expressed in this publication are those of the author(s) and not necessarily those of the funders.

## Author contributions

Conceptualization: F.C., E.M.H.; Data curation: M.T., F.C.; Formal bioinformatic analysis: F.C., M.M.; Funding acquisition: F.C., E.M.H., S.J.P.; Investigation: F.C., EM.H.; Bioinformatics methodology: F.C., M.M. and D.J.; Laboratory methodology: B.B., K.B. and E.M.H.; Project administration: E.M.H. and S.J.P.; Resources: J.P., E.M.H. and S.J.P.; Supervision: E.M.H., J.A.G., J.P. and S.J.P.; Results contextualization: F.C., E.M.H., R.C.M., T.W., and J.A.G. Validation: B.B., K.B. and R.C.M.; Visualization: FC; Writing – original draft: F.C. and E.M.H.; Writing – review & editing: all authors.

## Competing interests

The authors declare no competing interests.
