## [Transparent Peer Review file · Nature Communications]

The mutational landscape of *Staphylococcus aureus* during colonisation

Corresponding Author: Dr Francesc Coll

Version 0:

Reviewer comments:

Reviewer #1

(Remarks to the Author)

Very interesting manuscript addressing an important question of which genomic variants are associated with colonization of humans by *S. aureus*. The manuscript is well written, and data well presented. One concern I have with the manuscript is whether the experimental setup is ridged enough to really reveal colonization associated SNPs? The consideration is that the first sampling point most likely will be from a person who already has been colonized for an extended period of time and thus, the subsequent samplings from the same patient are “only” after longer periods of colonization. Thus, extended colonization is being assessed rather than variants associated with colonization per se. Also, the authors find that most diversity selected over time is in the *agr* quorum sensing locus which has been seen in several other studies comparing colonization with infection/persistent infection and *agr* mutations also arise during passaged in the laboratory. This diversity in conditions leading to *agr* diversity could be addressed in greater detail in the discussion. For the metabolic mutations these can also arise over time during passaged in the laboratory (the authors actually cites an excellent study of the topic in reference # 37) as so the authors should comment if there are any similarities in the changes observed in the present study compared to laboratory passaging and the other GWAS studies of colonization/persistence/infection. Lastly the riboflavin biosynthesis mutations could be addressed in greater detail as indicated below.

Abstract: The *agr* mutations should be mentioned in greater detail in the abstract as they also appear in many other GWAS studies. Also, a perspective on what the mutations found may mean in perspective of colonization.

L 37: The function role is not determined with respect to understanding colonization

L 61 : also mention gut

L 83-84: please provide a reference for the strain replacement statement

L 103: from which countries are the strain coming from – could be good to mention here

L 108-110: This is a surprisingly high frequency of MRSA – please comment

L 123: Is this common?

L 153: are the riboflavin mutations associated with altered resistance to roseoflavin? A section similar to the *nasD* could be addressing the finding of the *rib* genes and the role of these genes in vivo

L 205 and section reveals that there may have been significant antibiotic exposure which could have influenced the results. This point is clearly highlighted as a limitation by the authors but it complicated the interpretations of the data.

Discussion: Can be shortened but a more extensive comparison should be with the findings of other GWAS like studies such as those referenced in the manuscript as citations # 4, 6, 7, 10, 13, 16. The degree to which the mutations found in the present study are unique could add to support the uniqueness of this study compared to other studies of *S. aureus* interactions with the host.

L 330: Do you really have evidence of evolutionary pressures in the host? When *S. aureus* is passaged in the laboratory *agr* mutations frequently arise.

(Remarks on code availability)

Reviewer #2

(Remarks to the Author)

The paper by Coll et al represents a large population-level analysis of *S. aureus* from the nasopharynx. It is the largest study of its kind to date and represents a comprehensive analysis of genes that are under diversifying selection during human carriage. The major findings are the identification of the list of core genes that have an elevated frequency of amino acid altering mutations compared to the rest of the genome. These genes represent promising targets for investigating the biology of *S. aureus* during human carriage as highlighted by the authors.

The identification of mutations associated with resistance to antibiotics is of interest and consistent with selection during antibiotic exposure *in vivo*. The identification of potential novel resistance mutations is a useful contribution to the field.

The identification of agr mutants reinforces the known relevance of these mutations for *S. aureus* during carriage.

The identification of an enrichment for mutations in Nitrogen metabolic enzymes is intriguing. While the use of Tn-insertion mutants of the affected loci clearly indicates the relevance of those genes for utilising nitrogen from different sources, the analysis of the natural isolates with mutations compared to a parent strains are more informative. The authors discovered one mutation in nasD that correlated with increased growth in the presence of urea. However, the other 2 mutations examined gave equivalent or reduced growth in different nitrogen sources. Minor point- Without introducing the mutation into an isogenic background (or reversing the mutation), can the authors safely rule out the relevance of other mutations (or MGE) that may exist between the paired isolates ?

It is unclear from the discussion if the apparent selection on nitrogen metabolism is due to the existence of distinct nitrogen source availabilities between human individuals (ie that would occur after a transmission event. Is there anything known variation across human populations in this respect Alternatively, are there distinct sub-niches in the nasopharynx within individuals that could select for diversification of these pathways ? The authors could extend the relevant discussion here. Overall, the importance of the variation identified in these metabolic pathways is unclear.

The follow-up dataset was useful for increasing the power to identify additional genes with elevated aminoacid altering mutations. SraP was an interesting hit but no functional analysis was carried out -possibly due to unavailability of the relevant isolates.

One limitation as highlighted by the authors is the lack of analysis of variation in gene content or large indels in the analysis. These could impact on the examined phenotypes between related isolates from the same individual. Although, the nature of the short-read sequences employed may prevent the assembly of some MGE, an identification of unmapped genes should be feasible and could be informative.

The authors did not discuss the importance or relevance of immune selection in selecting for the diversity identified. It would be informative to discuss what impact if any this could have on the targets identified during nasal colonization.

The Methods are well-described and the analysis robust.

(Remarks on code availability)

Reviewer #3

(Remarks to the Author)

The paper entitled "The Mutation Landscape of Staphylococcus aureus during Colonization" by Coll and others applied genome-wide mutation enrichment combined with phenotypic experiments to explore the adaptations associated with *S. aureus* colonization. The paper is well-written and brings a lot of new and important contributions to the field. My main concern is related to a limited number of strains in which phenotypic analysis was generated and the ability of those results to support their conclusions. I also have minor questions/suggestions for improving the paper. The authors stated that nasD Cys452Ser mutation enhanced the growth of urea, however, only three isolates were tested. The authors recognized their sampling limitations but also used the results from these 3 isolates as one of the main findings as stated in the abstract "Further evidence of adaptation to nitrogen availability was revealed by enrichment of mutation in assimilatory nitrite reductase (nasD) and urease (ureG), including mutations that enhance growth with urea as the sole nitrogen source. The authors should maintain their conclusions tightened by the strong genome-wide analysis, which in fact found an enrichment for mutations in these genes. Still, it can't be further extrapolated more than that using only 3 isolates for phenotypic tests.

My minor comment/suggestion was related to homologous recombination. Why the authors did not further explore recombination and decide to exclude it from the work? Homologous recombination is an important source of adaptation in *S. aureus* The authors stated on Line 122 that recombination accounted for 23% of overall mutation. Exploring the recombination would also add another comprehensive level of information about the mutational landscape during colonization, mainly if the authors explore if the recombination events were ancestral or recent events - which could be potentially associated with the adaptation during the colonization process. Tools like fastGEAR (<https://pubmed.ncbi.nlm.nih.gov/28199698/>) could be applied to detect ancestral recombination and recent events.

(Remarks on code availability)

Version 1:

Reviewer comments:

Reviewer #1

(Remarks to the Author)

I have reviewed the adjustments to the manuscript and I am happy with the changes.

(Remarks on code availability)

Reviewer #2

(Remarks to the Author)

The authors have addressed the comments to my satisfaction.

(Remarks on code availability)

Reviewer #3

(Remarks to the Author)

This reviewer does not have any further comments. My previous comments were addressed and with the improvements made after the first round of review, the authors were able to strength their findings.

(Remarks on code availability)

REVIEWER COMMENTS

Reviewer #1 (Remarks to the Author):

Very interesting manuscript addressing an important question of which genomic variants are associated with colonization of humans by *S. aureus*. The manuscript is well written, and data well presented.

One concern I have with the manuscript is whether the experimental setup is ridged enough to really reveal colonization associated SNPs? The consideration is that the first sampling point most likely will be from a person who already has been colonized for an extended period of time and thus, the subsequent samplings from the same patient are “only” after longer periods of colonization. Thus, extended colonization is being assessed rather than variants associated with colonization per se.

We have acknowledged this limitation in the Discussion: “Finally, some of the mutational adaptations identified here could well represent adaptations of *S. aureus* strains to longer or extended periods of colonization, as those expected in persistent carriers, as these strains are more likely to be isolated over multiple sampling points (and hence to be included in this study).”

We have also avoided using the term “associated with” carriage or colonisation (previously mentioned only in Abstract line 33) as we have not conducted an association analysis per se but identified putative adaptive mutations in colonization *S. aureus* strains.

Also, the authors find that most diversity selected over time is in the *agr* quorum sensing locus which has been seen in several other studies comparing colonization with infection/persistent infection and *agr* mutations also arise during passaged in the laboratory. This diversity in conditions leading to *agr* diversity could be addressed in greater detail in the discussion.

We agree that this finding should be contextualised in the Discussion, considering the diversity of conditions that can lead to *agr* diversity in *S. aureus*. We have added in the Discussion: “It also well known that *agr* mutants arise frequently during the course of infections,⁴ and *in vitro* during post-isolation handling.⁴⁵” See comment below on further evidence of *agr* mutations arising during colonisation.

For the metabolic mutations these can also arise over time during passaged in the laboratory (the authors actually cite an excellent study of the topic in reference # 37) as so the authors should comment if there are any similarities in the changes observed in the present study compared to laboratory passaging and the other GWAS studies of colonization/persistence/infection.

We have added a new paragraph in the Discussion contextualising our results with those of previous within-host genomics studies:

“Compared to previous and similar studies reporting on *S. aureus* genetic adaptations, a recent study⁸ also observed enrichments of mutations in *agrA*, *agrC*, *pbp2* and *mprF* between colonisation isolates of the same host. Mutations in genes encoding enzymes of the tricarboxylic acid cycle (TCA), such as *sucA*⁸ and *fumC*⁸², have also been detected frequently in skin-adapted strains. When investigating infection-specific adaptations, a high frequency of mutations in *agr* genes is often reported,⁵ as well as in genes involved in antibiotic resistance (e.g., *rpoB* and *rpsJ*).^{8”}

Lastly the riboflavin biosynthesis mutations could be addressed in greater detail as indicated below.

See response to comment below.

Abstract: The *agr* mutations should be mentioned in greater detail in the abstract as they also appear in many other GWAS studies. Also, a perspective on what the mutations found may mean in perspective of colonization.

We have spelt out the identification of *agr* mutants in the Abstract to highlight this finding: “in regulators of quorum-sensing (*agrA* and *agrC*)” and “changes in haemolytic activity attributed to *Agr* mutants”.

The Discussion paragraph in lines 448-451 includes a perspective on what *Agr* mutations may mean in the context of *S. aureus* colonization. As mentioned above, we have also highlighted the diversity of known conditions that can lead to *agr* diversity in *S. aureus*.

L 37: The function role is not determined with respect to understanding colonization

We have edited this Abstract statement to indicate that: “We demonstrated the phenotypic effect of multiple adaptive mutations *in vitro*.”

L 61: also mention gut

We have edited the previous statement to indicate this: “bacterium may colonise other body sites including the skin, pharynx, axillae, perineum and the intestine.^{1,2”} and cited a recent review on the topic of *S. aureus* colonisation: 2. Piewngam, P. & Otto, M. *Staphylococcus aureus* colonisation and strategies for decolonisation. The Lancet Microbe 5, e606–e618 (2024).

L 83-84: please provide a reference for the strain replacement statement

We have provided the following reference: “Votintseva AA, Miller RR, Fung R, et al. Multiple-Strain Colonization in Nasal Carriers of *Staphylococcus aureus*. Carroll KC, ed. *Journal of Clinical Microbiology*. 2014;52(4):1192-1200. doi:10.1128/JCM.03254-13”

L 103: from which countries are the strain coming from – could be good to mention here

We have included information on countries of origin in the text for both the original dataset: “Most individuals (n=701, 88.6%) were sampled in the UK, followed by Singapore

(n=62, 7.8%), Thailand (n=17, 2.1%) and Ireland (n=11, 1.4%).” and for the additional dataset: “Most individuals (n=309, 42.3%) were sampled in the USA, followed by Singapore (n=186, 25.4%), UK (n=141, 19.3%), Japan (n=30, 4.1%), Thailand (n=26, 3.6%) and other countries (n=39, 5.3%).”

We have also added a new tab on “study information” in Supplementary Data 1 that includes the region and country of origin, setting and MRSA screening.

L 108-110: This is a surprisingly high frequency of MRSA – please comment

This is due to the inclusion of studies that only targeted and screened for MRSA, including 6 out the 10 studies included in the initial dataset (auguet2016, chow2017, coll2017, harrison2016, paterson2015 and tong2015). This high frequency of MRSA does not in any way represent a real epidemiological prevalence. We have made this point clear in this statement (line 118).

L 123: Is this common?

Previous studies assessing homologous recombination in *S. aureus* had already pointed to mobile genetic elements in *S. aureus* being hotspots of homologous recombination: Everitt, R. G. *et al.* Mobile elements drive recombination hotspots in the core genome of *Staphylococcus aureus*. *Nat. Commun.* 5:3956 doi: 10.1038/ncomms4956 (2014).

We have added the following statement in lines 143-145: “This is consistent with previous studies reporting that most homologous recombination in the core genome of *S. aureus* can be found at or around mobile genetic elements.”

L 153: are the riboflavin mutations associated with altered resistance to roseoflavin? A section similar to the nasD could be addressing the finding of the rib genes and the role of these genes in vivo.

We thank this reviewer for proposing this hypothesis. We could not follow up riboflavin mutations as we did not have available natural mutants of both *ribD* and *ribA* (the two most mutated genes in the riboflavin biosynthesis pathway) from internal collections that we could reculture and follow up experimentally. Still, we have added a statement in the Discussion postulating this hypothesis: “Riboflavin biosynthesis was the second and eighth most mutated metabolic pathway in the initial and extended datasets, respectively, with *ribD* and *ribA* being the two most mutated genes in this pathway (Supplementary Data 2). We hypothesise that mutations in these genes might result in altered resistance to roseoflavin, an antimetabolite analogue of riboflavin that has antimicrobial properties,⁶⁹ but could not test this hypothesis as we lacked natural mutants of *ribD* and *ribA* from internal collections.” (lines 392-397 in the Discussion)

L 205 and section reveals that there may have been significant antibiotic exposure which could have influenced the results. This point is clearly highlighted as a limitation by the authors but it complicated the interpretations of the data.

The point we raised as a limitation is the fact that “most [colonization] isolates came from studies of *S. aureus* colonisation in hospitalised patients”, which are more likely to have been treated with antibiotics. Availability of antibiotic treatment data would have facilitated the interpretation of the findings, by linking antibiotic usage to acquisition of adaptive AMR mutations. Still, we could link AMR mutations with reduced susceptibility to their cognate antibiotics *in vitro* (Supplementary Figure 6).

Discussion: Can be shortened but a more extensive comparison should be with the findings of other GWAS like studies such as those referenced in the manuscript as citations # 4, 6, 7, 10, 13, 16. The degree to which the mutations found in the present study are unique could add to support the uniqueness of this study compared to other studies of *S. aureus* interactions with the host.

See comment above about the contextualisation of our studies with similar genomic studies.

L 330: Do you really have evidence of evolutionary pressures in the host? When *S. aureus* is passaged in the laboratory *agr* mutations frequently arise.

We have now acknowledged that some of the *Agr* mutants identified could have arisen during laboratory passage. To provide evidence of *Agr* mutants arising *in host*, we focused on individuals with more than one isolate carrying the same *Agr* mutation isolated from independent swabs. We have added the following text in the Results (lines 304-310): “Because *agr* mutants could also have arisen during laboratory passage,⁴⁸ we sought to provide evidence of *agr* mutants arising *in host*. The identification of the same mutation in the same strain from independent swabs (i.e., taken at different time points) would point to a single origin of such mutation arising *in host*, as it is less likely for the same mutation to originate multiple independent times in the laboratory. Indeed, we found that all five individuals with multiple *Agr*-mutated isolates taken from independent swabs (Supplementary Table 7) carried the same *Agr* mutation.”

Reviewer #2 (Remarks to the Author):

The paper by Coll et al represents a large population-level analysis of *S. aureus* from the nasopharynx. It is the largest study of its kind to date and represents a comprehensive analysis of genes that are under diversifying selection during human carriage. The major findings are the identification of the list of core genes that have an elevated frequency of amino acid altering mutations compared to the rest of the genome. These genes represent promising targets for investigating the biology of *S. aureus* during human carriage as highlighted by the authors.

We thank this reviewer for highlighting the strengths of our study as well as the limitations, see responses to individual comments below.

The identification of mutations associated with resistance to antibiotics is of interest and consistent with selection during antibiotic exposure *in vivo*. The identification of potential novel resistance mutations is a useful contribution to the field.

The identification of agr mutants reinforces the known relevance of these mutations for *S. aureus* during carriage.

The identification of an enrichment for mutations in Nitrogen metabolic enzymes is intriguing. While the use of Tn-insertion mutants of the affected loci clearly indicates the relevance of those genes for utilising nitrogen from different sources, the analysis of the natural isolates with mutations compared to a parent strains are more informative. The authors discovered one mutation in *nasD* that correlated with increased growth in the presence of urea. However, the other 2 mutations examined gave equivalent or reduced growth in different nitrogen sources. Minor point- Without introducing the mutation into an isogenic background (or reversing the mutation), can the authors safely rule out the relevance of other mutations (or MGE) that may exist between the paired isolates?

We agree that generating and testing clean isogenic mutants would have provided stronger and more conclusive evidence about the role of individual mutations. As suggested, we now report all mutations and large indels as identified by Breseq between the pairs of related isolates from the same individual that were tested *in vitro* (Supplementary Data 5).

It is unclear from the discussion if the apparent selection on nitrogen metabolism is due to the existence of distinct nitrogen source availabilities between human individuals (ie that would occur after a transmission event. Is there anything known variation across human populations in this respect Alternatively, are there distinct sub-niches in the nasopharynx within individuals that could select for diversification of these pathways? The authors could extend the relevant discussion here. Overall, the importance of the variation identified in these metabolic pathways is unclear.

We thank this reviewer for suggesting this possibility. As far as we are aware, only one study characterised the metabolite composition of nasal secretions, only from eight different individuals (DOI: 10.1371/journal.ppat.1003862), but not that of other typical

colonization sites such as the nasopharynx. Other studies have focused on quantifying specific metabolites such as lipids (DOI: 10.1038/s41598-021-93817-1) in human noses.

We have expanded the Discussion to discuss this point (lines 382-390): “Adaptions in nitrogen metabolic genes could be a response to distinct nitrogen source availabilities between human hosts, variation in the community structure of the nasal microbiota or between distinct sub-niches within the nasopharynx, therefore requiring metabolic adaptation of *S. aureus* strains when transmitting onto a new host or move to another anatomical niche. All in all, the specific phenotypic and physiological importance of variation in nitrogen metabolic genes is unclear and requires further characterisation.”

The follow-up dataset was useful for increasing the power to identify additional genes with elevated amino acid altering mutations. SraP was an interesting hit but no functional analysis was carried out -possibly due to unavailability of the relevant isolates.

We focused our follow-up phenotypic characterisation on the hits originally identified in the initial dataset, not those later identified in the extended dataset. This is why SraP mutations were not functionally characterised. Still, we found it relevant to map the position of amino acid substitutions on the protein 3D structure which revealed that many substitutions map onto the L-lectin domain and have thus the potential to directly impact ligand binding. Co-authors of this study are currently following up these findings experimentally.

One limitation as highlighted by the authors is the lack of analysis of variation in gene content or large indels in the analysis. These could impact on the examined phenotypes between related isolates from the same individual. Although, the nature of the short-read sequences employed may prevent the assembly of some MGE, an identification of unmapped genes should be feasible and could be informative.

We have now reported all mutations and large indels as identified by Breseq between the pairs of related isolates from the same individual that were tested *in vitro* (Supplementary Data 5).

The authors did not discuss the importance or relevance of immune selection in selecting for the diversity identified. It would be informative to discuss what impact if any this could have on the targets identified during nasal colonization.

We have included a new paragraph in the Discussion that discuss the possible role of some of the mutational adaptions in immune evasion: “As previously proposed,⁸ adaptive antibiotic resistance mutations could result in other phenotypic effects (pleiotropic effects) or be instead the result of immune evasion adaptations. As such, *mprF* mutations reduce susceptibility to cationic antibiotics like daptomycin⁸⁰ but also to host defence cationic peptides produced by neutrophils,⁸¹ representing a potential immune evasion strategy. Similarly, mutations in *tarS*, which encodes the wall teichoic acid (WTA) β -glycosyltransferase and was frequently mutated in the extended dataset, could be adaptive in the context of immune evasion, as it has been shown that *tarS* knockout mutants escape host anti-WTA-IgG mediated opsonophagocytosis.⁷⁷ *tarS* variants could

also be influencing susceptibility to staphylococcal phages and beta-lactams, as TarS-mediated WTA β -glycosylation is required for susceptibility to podoviruses⁷⁸ and methicillin resistance, respectively.⁷⁹"

The Methods are well-described and the analysis robust.

Reviewer #3 (Remarks to the Author):

The paper entitled "The Mutation Landscape of *Staphylococcus aureus* during Colonization" by Coll and others applied genome-wide mutation enrichment combined with phenotypic experiments to explore the adaptations associated with *S. aureus* colonization. The paper is well-written and brings a lot of new and important contributions to the field. My main concern is related to a limited number of strains in which phenotypic analysis was generated and the ability of those results to support their conclusions. I also have minor questions/suggestions for improving the paper.

We thank this reviewer for highlighting the many contributions of our study to the field, and pointing to limitations that need addressing. See our comments below.

The authors stated that *nasD* Cys452Ser mutation enhanced the growth of urea, however, only three isolates were tested. The authors recognized their sampling limitations but also used the results from these 3 isolates as one of the main findings as stated in the abstract "Further evidence of adaptation to nitrogen availability was revealed by enrichment of mutation in assimilatory nitrite reductase (*nasD*) and urease (*ureG*), including mutations that enhance growth with urea as the sole nitrogen source. The authors should maintain their conclusions tightened by the strong genome-wide analysis, which in fact found an enrichment for mutations in these genes. Still, it can't be further extrapolated more than that using only 3 isolates for phenotypic tests.

We have rephrased the Abstract statements related to nitrogen metabolism to tighten our conclusions to our genomic and phenotypic observations. Previously we had indicated that:

"Nitrogen metabolism and riboflavin synthesis were the metabolic processes with strongest evidence of adaptation. Further evidence of adaptation to nitrogen availability was revealed by enrichment of mutations in the assimilatory nitrite reductase (*nasD*) and urease (*ureG*), including mutations that enhance growth with urea as the sole nitrogen source."

Now we indicate:

"Nitrogen metabolism was the metabolic process with the strongest evidence of adaptation, with the assimilatory nitrite reductase (*nasD*) and urease (*ureG*) showing the highest enrichment of mutations. We identified a *nasD* natural mutant with enhanced growth under urea as the sole nitrogen source."

Also note that we have now removed “riboflavin synthesis” from the Abstract because, as indicated in the response to a comment from reviewer #1: “We could not follow up riboflavin mutations as we did not have available natural mutants of both *ribD* and *ribA* (the two most mutated genes in the riboflavin biosynthesis pathways) from internal collections that we could reculture and follow up experimentally.”

My minor comment/suggestion was related to homologous recombination. Why the authors did not further explore recombination and decide to exclude it from the work? Homologous recombination is an important source of adaptation in *S. aureus*. The authors stated on Line 122 that recombination accounted for 23% of overall mutation. Exploring the recombination would also add another comprehensive level of information about the mutational landscape during colonization, mainly if the authors explore if the recombination events were ancestral or recent events - which could be potentially associated with the adaptation during the colonization process. Tools like fastGEAR (<https://pubmed.ncbi.nlm.nih.gov/28199698/>) could be applied to detect ancestral recombination and recent events.

We excluded homologous recombination because although it accounted for 23% of the overall mutation count, most recombination (80%) fell within three prophage regions (Supplementary Figure 3). This leaves only 4.6% of the overall mutation count attributable to recombination outside prophage regions. This observation is consistent with previous studies assessing homologous recombination in *S. aureus*, which had pointed to mobile genetic elements being hotspots of homologous recombination in the core genome of *S. aureus* (Everitt *et al.* 2014, doi: 10.1038/ncomms4956).

These results may point to a translocation and movement of phages during carriage which, given the highly mosaic structure of phages, would need to be resolved using long-read sequencing. The recombination and movement of Staphylococcal phages, and possible adaptive effect of these changes during *S. aureus* colonization, should be the focus of future studies.

We have highlighted this point in the “Limitations” paragraph of the Discussion (lines 481-486): “We did not explore the role of homologous recombination in adaptation. Of note, 80% of mutations attributable to recombination fell within three prophage regions, consistent with previous studies showing that MGEs are hotspots of homologous recombination in the core genome of *S. aureus*.³² These results point to a translocation and movement of Staphylococcal phages during colonization, the adaptive effect of which should be the focus of future studies.”